# Nuclear IGF1R interact with PCNA to preserve DNA replication after DNA-damage in a variety of human cancers

**Chen Yang**[1,2,3]**, Yifan Zhang**[1]**, Yi Chen**[1]**, Franziska Ragaller**[1,4]**, Mingzhi Liu**[1]**, Sara Corvigno**[1]**, Hanna Dahlstrand**[1,5]**, Joseph Carlson**[1]**, Zihua Chen**[2,3]**, Anders Näsman**[1]**, Ahmed Waraky**[6]**, Yingbo Lin**[1]**, Olle Larsson**[1]**, Felix Haglund**[1]*

**1** Department of Oncology and Pathology, Karolinska Institutet, Stockholm, Sweden, **2** Department of General Surgery, Xiangya Hospital of Central South University, Changsha, Hunan, China, **3** Hunan Key Laboratory of Precise Diagnosis and Treatment of Gastrointestinal Tumor, Changsha, Hunan, China, **4** Heidelberg University and German Cancer Research Center (DKFZ), Heidelberg, Germany, **5** Department of Immunology, Genetics and Pathology, Uppsala University, Uppsala, Sweden, **6** Department of Laboratory Medicine, Gothenburg University, Gothenburg, Sweden

* Felix.Haglund@ki.se

**Data Availability Statement:** The data underlying the results presented in the study are available

## Abstract

Nuclear IGF1R has been linked to poor outcome in cancer. We recently showed that nuclear IGF1R phosphorylates PCNA and increases DNA damage tolerance. In this paper we aimed to describe this mechanism in cancer tissue as well as in cancer cell lines. *In situ* proximity ligation assay identified frequent IGF1R and PCNA colocalization in many cancer types. IGF1R/PCNA colocalization was more frequently increased in tumor cells than in adjacent normal, and more prominent in areas with dysplasia and invasion. However, the interaction was often lost in tumors with poor response to neoadjuvant treatment and most metastatic lesions. In two independent cohorts of serous ovarian carcinomas and oropharyngeal squamous cell carcinomas, stronger IGF1R/PCNA colocalization was significantly associated with a higher overall survival. *Ex vivo* irradiation of ovarian cancer tissue acutely induced IGF1R/PCNA colocalization together with γH2AX-foci formations. *In vitro*, RAD18 mediated mono-ubiquitination of PCNA during replication stress was dependent on IGF1R kinase activity. DNA fiber analysis revealed that IGF1R activation could rescue stalled DNA replication forks, but only in cancer cells with baseline IGF1R/PCNA interaction. We believe that the IGF1R/PCNA interaction is a basic cellular mechanism to increase DNA stress tolerance during proliferation, but that this mechanism is lost with tumor progression in conjunction with accumulated DNA damage and aberrant strategies to tolerate genomic instability. To exploit this mechanism in IGF1R targeted therapy, IGF1R inhibitors should be explored in the context of concomitant induction of DNA replication stress as well as in earlier clinical stages than previously tried.

from https://figshare.com/ with a DOI: 10.6084/
m9.figshare.12388256.

**Funding:** This work is supported by the Swedish
Cancer Society, the Swedish Childhood Cancer
Foundation, The Cancer Society in Stockholm, the
Stockholm County Council and the China
Scholarship Council [CSC201706370014] and
Karolinska Institutet.

**Competing interests:** The authors have declared
that no competing interests exist.

## Introduction

During cell division numerous events may hinder the accurate and complete replication of the
genome, commonly referred to as replication stress. Increased replication stress may induce
replication fork stalling and ultimately fork collapse [1]. Normal cells have the capacity to over-
come replication stress, with the aid of DNA damage response mechanisms. In cancer, faulty
DNA repair systems (including mismatch repair, recombinational repair, nucleotide excision
repair or base excision repair) frequently result in accelerated genome destabilization [2]. To
overcome this complication, cancer cells may utilize DNA maintenance protocols to guarantee
genomic replication in face of endogenous and exogenous replication stress [3, 4]. The DNA
damage tolerance (DDT) pathway has recently been recognized as a mechanism to overcome
replication stress induced fork stalling either by error-prone translesion synthesis (TLS) or
error-free template switching (TS) [5, 6]. A key mediator of DDT is the DNA clamp proliferat-
ing cell nuclear antigen (PCNA) which functions by recruiting proteins to the DNA.

Our group has recently shown that nuclear insulin-like growth factor 1 receptor (nIGF1R)
directly phosphorylates three PCNA tyrosine residues (Tyr-60, -133, and -250), resulting in
recruitment of ubiquitination E2/3 ligases, PCNA mono- and poly-ubiquitination and replica-
tion fork progression after DNA damage in human embryonic stem cells [7]. It is yet to be
determined if this mechanism is present in cancer cells.

Previous studies have linked nIGF1R to other recombinational DNA repair pathways
(homologous repair (HR) and non-homologous end joining (NHEJ)), and IGF1R inhibition
has been shown to decrease HR and delay the resolution of γH2AX foci (in a fashion similar to
BRCA1/2 deficient tumors) [8–13]. In cancer, tumor expression of nIGF1R has been linked to
chemotherapy and radiotherapy resistance [14–16].

In this study we aimed to explore and describe the potential interaction between IGF1R
and PCNA in cancer tissue and cancer cell lines. First, we developed a protocol to identify
IGF1R/PCNA colocalization in tissue from clinical tumors. After establishing that most inves-
tigated tumor subtypes exhibited IGF1R/PCNA colocalization, we investigated how the inter-
action was affected by irradiation in ovarian cancer tissue *ex vivo*. To assess its clinical
relevance of the colocalization, we investigated two independent retrospective cancer cohorts.
Finally, we investigated the function of the IGF1R/PCNA interaction in several cancer cell
lines.

## Material and methods

### Patient samples and ethical permission

The use of patient tumor and normal tissue was approved by the local ethics committee
(Regionala Etikprövningsnämnden Stockholm) through two permissions (Registration num-
bers 2012/539-31/1 and 2017-1035-31/2). The use of anonymized tissue samples is in accor-
dance with the Swedish biobank law. All patients gave oral and written consent before
prospective sample collection. All methods were carried out in accordance with relevant guide-
lines (Declaration of Helsinki) and regulations.

### Immunohistochemistry and immunofluorescence

See Supplementary Material and Methods in S1 Text. Representative microphotographs are
shown in S1 Fig (IGF1R in cells), S2 Fig (IGF1R and PCNA in control tissues), S3 Fig (cancer
tissue) and S11 Fig (DNA Polymerase ETA foci).

### *In situ* Proximity Ligation Assay (PLA)

See Supplementary Material and Methods in S1 Text. Representative microphotographs of IGF1R/PCNA PLA validation are shown in S4 Fig (FFPE tissue, brown dots indicating colocalization of the two proteins) and Fig 5D (Cells, red fluorescence dots indicating colocalization of the two proteins).

### Tissue microarray scoring

The tissue microarrays (TMA) were scored by a clinical pathologist blinded to clinical outcome. Total and nuclear PLA signals were evaluated for both IGF-1R/PCNA and Rad18/PCNA. Tumors were arbitrarily classified for statistical comparisons: tumors with no or very few signals were scored as +1 (negative / weak); tumors with moderate signals (5–10 per cell/nuclei in the majority of cells) were scored as +2 (intermediate), and tumors with abundant signals (>10 signals per cell/nuclei in the majority of cells) were scored as +3 (strong).

In addition, the prevalence of Rad18/PCNA signal clusters was rounded off to the nearest 5% and after reviewing the cases arbitrary cutoffs were determined: Tumors with very few signal clusters estimated at <1% were scored as +1; tumors with an estimated 2–50% prevalence of signal clusters were scored as +2 and tumors with an estimated 51–100% prevalence of signal clusters were scored as +3.

### *Ex vivo* irradiation of tumor samples

Anonymized tumor and normal samples used for *ex vivo* irradiation experiments were obtained from the Department of Clinical Pathology, Karolinska Hospital, Sweden. The tissue was isolated during gross examination by a clinical pathologist. Directly after a patient's surgery the tissue was stored in DMEM (10% FBS) and kept at 4° C until the start of treatment (within 2 hours). The tissue of each patient was divided into three parts, which were then treated with different doses of X-Ray irradiation (0, 2 and 8 Grey). After treatment the tissue was incubated in DMEM (4.5 g/L Glucose and 10% FBS) for 1 hour before fixation in formalin for 24 hours. After fixation the samples were transferred into 70% Ethanol and subsequently embedded in paraffin. The tissue was cut in 4 μm thick sections for *in situ* PLA.

### Cell lines and reagents

See Supplementary Material and Methods in S1 Text.

### Immunoblotting and immunoprecipitation

See Supplementary Material and Methods in S1 Text.

### DNA fiber assay

To investigate the DNA replication fork dynamics, DNA molecules were pulse-labeled with 50μM halogenated nucleotides CIdU (C6891, Sigma) and 500μM IdU (I7125, Sigma) for 20 min respectively before and after 0.2mM HU treatment. IGF1, NVP and IGF1+ NVP treatments were carried out for 60 min before the HU treatments (Fig 5A top panel). 20 min after HU treatment, cells were harvested and diluted to $10^6$ cells/ml. Spreading and staining were performed as described previously [7]. CIdU and IdU integrated in the DNA molecules were immunostained with monoclonal rat anti-BrdU (Clone BU1/175, Bio-Rad) and monoclonal mouse anti-BrdU (Clone B44, Becton Dickinson, 347580) respectively and visualized by anti-rat AlexaFluor594 (Molecular probes) and anti-mouse AlexaFluor488 (Molecular Probes) secondary antibodies respectively. Fluorescence images were captured

using AxioImager M2 fluorescence microscope (Zeiss) and analyzed using the ImageJ software. At least 50 unidirectional forks labeled with both CIdU and IdU were measured for every condition. The elongation ratio was calculated by length of IdU/length of CIdU, quantitatively reflecting the relative changes in replication fork efficiency.

### Statistical analyses

For DNA fiber assay, statistical analysis was performed using R (version 3.6.0) (R Core Team (2013). We used the Student t-test to compare between treatment groups. The differences were considered significant at $p < 0.05$. Violin plots were drawn by R Studio (RStudio Team (2015)), code was uploaded to GitHub (https://github.com/Minliuv18/DNAFIBERASSAY.git).

Data from the clinical cohorts were analyzed using SPSS version 24 (IBM software). Overall survival was compared with Kaplan-Meier estimates and differences were tested with log-rank comparisons. Numerical and categorical variables were compared with Spearman's ranked correlation, Fishers' exact tests or Mann-Whitney U as applicable.

## Results

### IGF1R and PCNA interaction in a clinical cancer panel

Prior to investigating tumor specimens, the specificity and concentrations of the utilized antibodies were validated on cells and tissue with known expression of the target proteins (S1 and S2 Figs). The *in-situ* PLA method was validated in tissue samples with confirmed immunoreactivity (S3 and S4 Figs) as described in the Supplementary Material and Methods in S1 Text.

To study the activation of the DDT pathway in cancer tissue, we performed *in situ* PLA for IGF1R/PCNA colocalization in 35 tumors (29 primary and 6 metastases) from 30 patients (Fig 1). The panel included ovarian, mammary, colorectal and pulmonary adenocarcinomas, urothelial carcinomas, malignant melanomas, squamous cell carcinomas, one Merkel cell carcinoma, one sarcoma and one parathyroid adenoma as described in S1 Table.

Both nuclear and cytoplasmic IGF1R/PCNA colocalization signals were observed at varying levels in tumor tissue. Perhaps most strikingly, we observed a quite frequent and clear increase of nuclear and cytoplasmic PLA signals in tumor cells as compared to the adjacent normal tissue, tumor infiltrating lymphocytes and tumor stroma cells (fibroblasts and myofibroblasts). The nuclear colocalization signals were semi-quantified in the primary tumors as moderate (>5 signals per nuclei in >30% of the tumor cells; 11/28 tumors), few (1–4 signals per nuclei; 9/28 tumors) or no/very few signals (0–1 signals per nuclei; 6/28 tumors). Correspondingly, the metastases had few (n = 4) or null (n = 2) signals. Cases with more nuclear signals also had more cytoplasmic signals (Fishers' exact test, $p < 0.001$). Among the cases that stained the strongest were cases of malignant melanoma, high-grade ovarian carcinoma (HGSC), invasive urothelial carcinoma, pulmonary carcinomas, ductal mammary adenocarcinoma and squamous cell carcinomas (SCC). In some cases, we saw heterogeneities within the tumor; a sarcoma with increased signals in cells with increased atypia (Fig 1) and cases of squamous cell carcinomas that had increased numbers of signals at the invasive front (S5 Fig). Some tumors, including a Merkel cell carcinoma (Fig 1), a cervical squamous cell carcinoma, a parathyroid adenoma and a lobular mammary carcinoma showed no colocalization signal in the tumor cells. There was no clear association between IGF1R/PCNA colocalization and tumor proliferation as determined by mitotic count (data not shown). These results indicate that the IGF1R/PCNA colocalization is upregulated in a subset of clinical cancers.

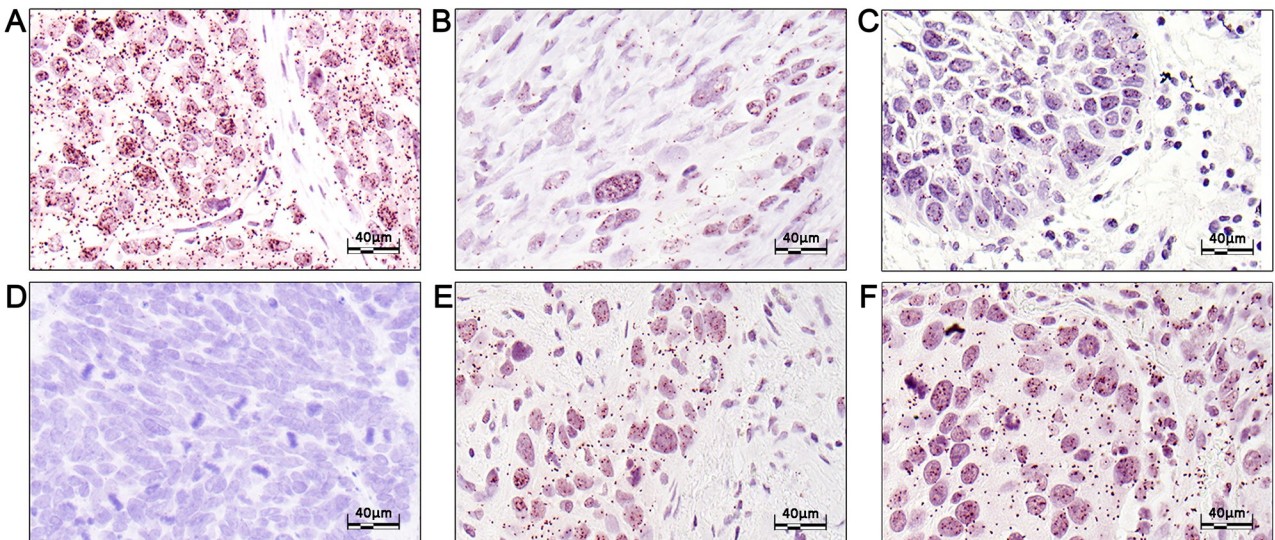

**Fig 1.** *In situ* **PLA showing IGF1R/PCNA colocalization in cancer tissue.** Representative photomicrographs of IGF1R/PCNA *in situ* proximity ligation assay (PLA) in different cancer forms. A) desmoplastic melanoma, B) malignant peripheral nerve sheath tumor, C) oropharyngeal squamous cell carcinoma, D) Merkel cell carcinoma, E) high-grade urothelial carcinoma, F) pulmonary adenocarcinoma. Strong nuclear and cytoplasmic colocalization was found in a desmoplastic melanoma (A). Sometimes, we observed more PLA signals in dysplastic cells, here represented by a malignant peripheral nerve sheath tumor (MPNST)/sarcoma (B). Varying levels of IGF1R/PCNA colocalization were found depending on the type of tumor, but it was frequently stronger in tumor cells than surrounding tumor stroma (E) or tumor infiltrating lymphocytes (C). Some cases showed no signals in the tumor cells, here represented by a Merkel cell carcinoma (D).

We compared the level of IGF1R/PCNA signals with the overall mutational burden (by ranking, S1 Table) of the primary tumors (without neoadjuvant treatment) as determined by the tumor subgroup as described by Chalmers et al. [17]. We were unable to find a significant correlation between mutational burden rank and IGF1R/PCNA colocalization (Spearman's' ranked correlation, $p = 0.176$). Given the large variation of mutational burden within tumor groups this test would have a high risk of a type II error.

Mono-ubiquitination of phosphorylated PCNA is mediated by E3 ligases, including Rad18. Following the observation of an increased IGF1R/PCNA colocalization in clinical tumors, we performed a similar PLA staining for Rad18 and PCNA on a subset of the clinical cancer panel (n = 16). The Rad18/PCNA PLA signals sometimes appeared as massive signal clusters (>50 signals per nuclei) in individual tumor cells. By using a semi-quantification similar to that for IGF1R/PCNA, we found a significant correlation between IGF1R/PCNA colocalization and Rad18/PCNA colocalization (Spearman's ranked correlation, r = 0.599, $p = 0.017$). With the exception of two neoadjuvantly treated mammary carcinomas, PLA showed moderate to strong Rad18/PCNA colocalization signals in all cases (including some IGF1R/PCNA negative cases). This suggests that E3-dependent PCNA mono-ubiquitination may be initiated by IGF1R independent mechanisms in clinical tumors, in line with functional experiments from other model organisms lacking *igf1r* [18, 19].

IGF1R/PCNA colocalization signals were also seen in normal tissue with high proliferation, including colon enterocytes in the basal part of the crypts and squamous cells in the skin and oral mucosa basal layers (*stratum basale*). Interestingly, in dysplastic or sun damaged squamous epithelium, we observed an increased number of signals which also included the prickle layer (*stratum spinosum*) as shown in S6 and S7 Figs.

## IGF1R/PCNA colocalization may be associated with clinical response to radio-chemotherapy

The analyzed cancer panel included a small series of neoadjuvantly treated breast carcinomas (n = 6) with poor chemotherapy response as determined by histopathology, two of which had paired pre- and post-therapy biopsies. In the post-chemotherapy resections 5/6 showed absence of IGF1R/PCNA colocalization signals and varying levels of RAD18/PCNA signals (strong in 3/5 cases, absent or few in 2/5). In one case of ductal adenocarcinoma that had received neoadjuvant treatment, the levels of IGF1R/PCNA signals went from moderate to none with an apparent decrease in nuclear IGF1R as determined by IHC (Fig 2). Additionally, one neoadjuvantly treated urothelial carcinoma was completely negative (as compared to two untreated cases with moderate IGF1R/PCNA signals). Since these 5/6 randomly selected chemotherapy resistant mammary carcinomas and a single neoadjuvantly treated urothelial carcinoma exhibited no IGF1R/PCNA interaction in the post-treatment setting, it is implied that the IGF1R/PCNA negative tumor cells survived radio-chemotherapy (selection) or that chemotherapy resulted in the downregulation of IGF1R/PCNA interaction (modification).

## IGF1R/PCNA colocalization is significantly associated with better overall survival in ovarian HGSC and oropharyngeal SCC

Given the variation of IGF1R/PCNA colocalization within single tumor subtypes and the putative response to radio- and chemotherapy, we conducted extended studies of two tumor subtypes that had among the strongest IGF1R/PCNA colocalization signals in the clinical cancer panel–invasive oropharyngeal SCC (OPSCC) and high-grade ovarian cancer (HGSC). To this end full tissue tumor slides from 23 patients with OPSCC and tumor tissue microarrays (TMA) including 238 patients with paired primary and metastatic HGSC were investigated for IGF1R/PCNA and Rad18/PCNA colocalizations.

In ovarian cancer, strong IGF1R/PCNA colocalization was significantly associated with better disease-specific overall survival (OS) (Fig 3, log rank, $p = 0.037$). This association was even stronger among patients with complete response after primary treatment (log rank, $p = 0.005$). While Rad18/PCNA signal intensity separated the Kaplan-Meier survival estimates, the differences were not statistically significant (log rank, $p = 0.190$ for overall staining intensity and $p = 0.223$ for signal clusters). There was a significant association between IGF1R/PCNA and Rad18/PCNA signal intensity ($p = 0.039$) and clusters ($p = 0.041$) in the primary (adnexal) tumor, but not in the metastatic lesions ($p = 0.125$ and $p = 0.263$) suggesting that HGSC metastases may have undergone alterations of the DNA damage tolerance pathway such as activating IGF1R independent PCNA phosphorylation.

Similarly, strong IGF1R/PCNA colocalization was significantly associated with better OS in the smaller OPSCC cohort (Fig 3, log rank, $p = 0.027$), and we saw no significant separation of the Kaplan-Meier survival estimates for Rad18/PCNA signals (log rank, $p = 0.826$ for overall intensity and $p = 0.214$ for clusters). There was no association with p16 overexpression by immunohistochemistry (>70%) or HPV DNA status in the OPSCC cohort ($p > 0.7$).

## Irradiation induces IGF1R/PCNA colocalization in ovarian cancer tissue *ex vivo*

To investigate the putative relationship between IGF1R/PCNA colocalization and replication stress *ex vivo*, fresh tissue was isolated directly after surgery from high-grade serous (n = 5) and mucinous (n = 1) ovarian carcinomas (schematically shown in Fig 4A).

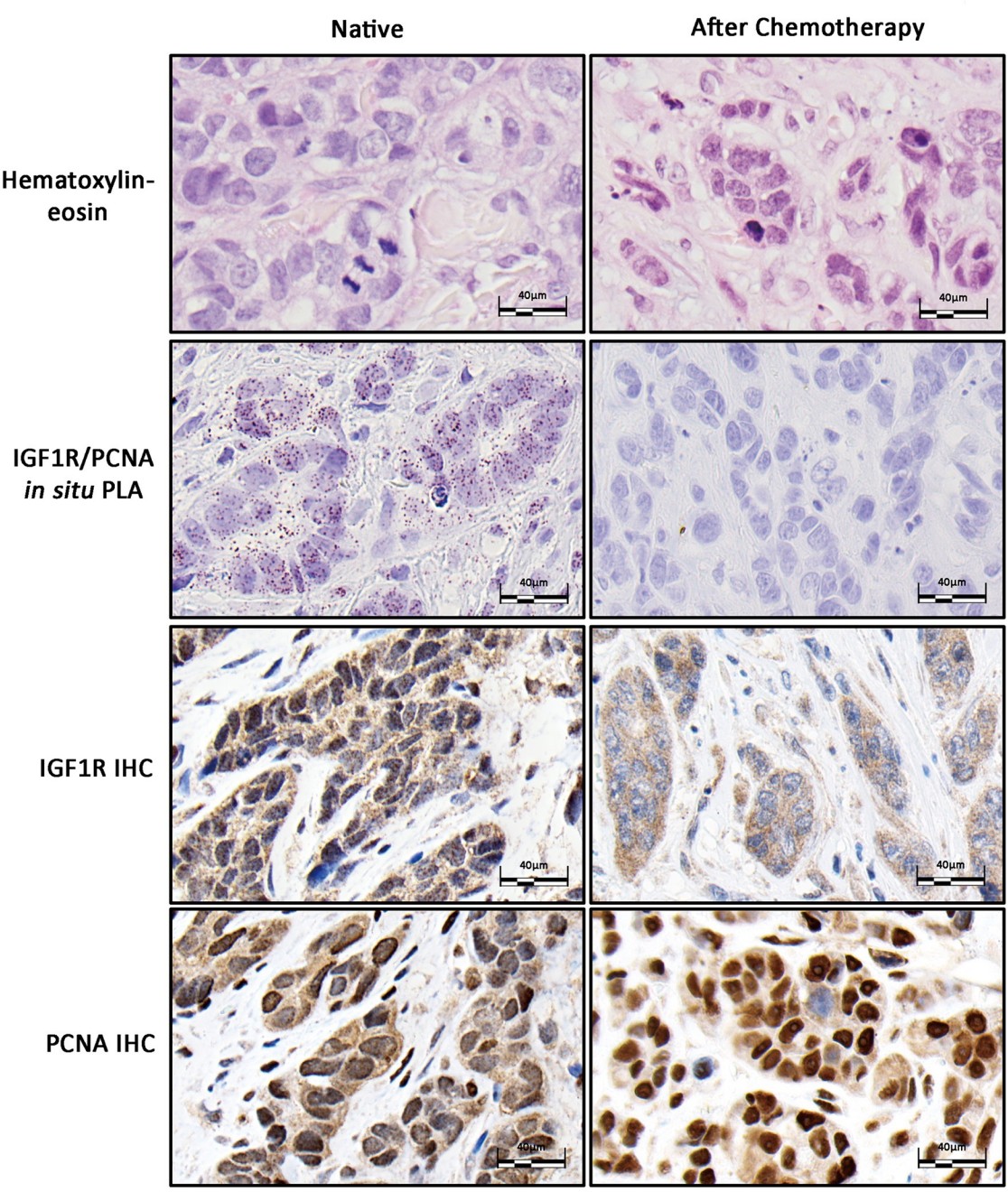

**Fig 2. *In situ* PLA and immunohistochemistry of IGF1R and PCNA in a neoadjuvantly treated breast carcinoma.** Representative microphotographs of a neoadjuvantly treated ductal type breast cancer (no special type, Nottingham grade 3). The preoperative biopsy (left) showed nuclear immunoreactivity of IGF1R as well as IGF1R/PCNA colocalization by PLA. In the surgical specimen (after neoadjuvant chemotherapy) the viable tumor cells showed intact mitotic rate with an apparent increase in PCNA immunoreactivity. The treated tumor cells showed a marked decrease of nuclear IGF1R-immunoreactivity and a total loss of IGF1R/PCNA colocalization signals.

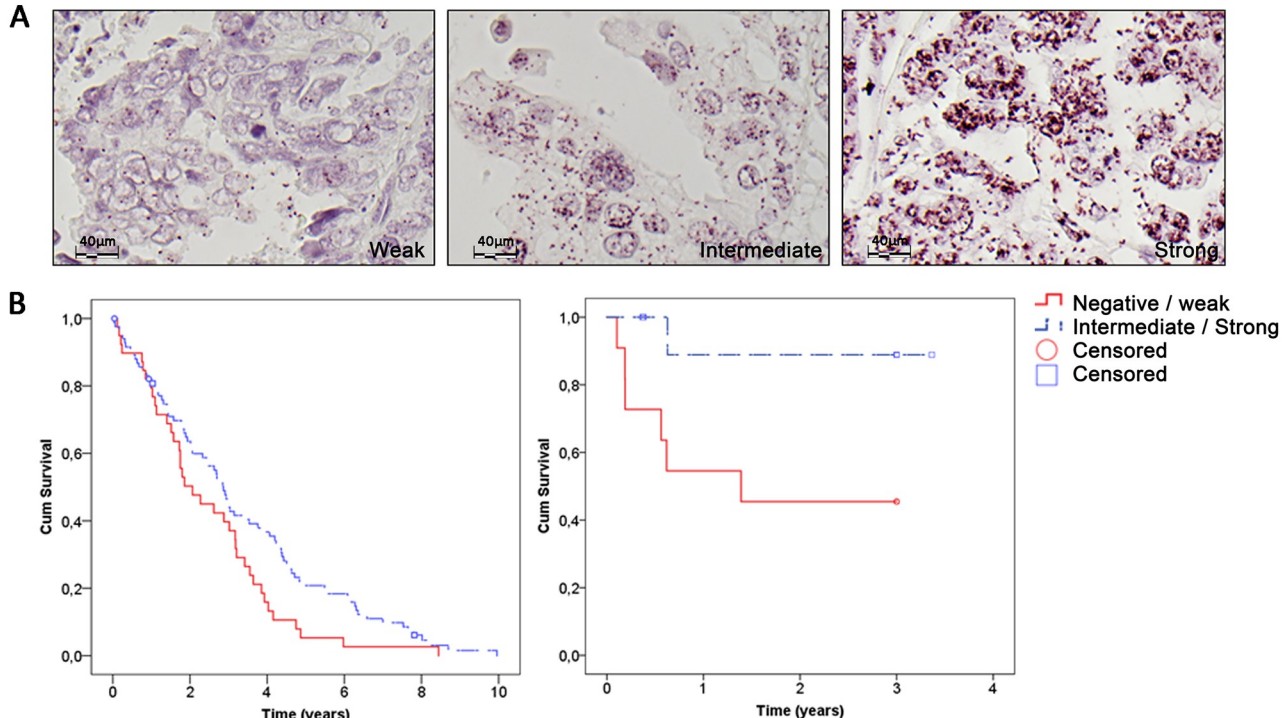

**Fig 3. IGF1R/PCNA colocalization is associated with better survival in high-grade ovarian and oropharyngeal squamous cell carcinomas.** A) Examples of scoring intensities of ovarian high-grade serous carcinomas (HGSC) in the tissue microarray (from left to right: weak, intermediate and strong). When comparing the survival of patients, the tumors' staining intensity was clustered as negative/weak (score: +1) or intermediate/strong (score: +2/+3). B) Higher IGF1R/PCNA colocalization was associated with better overall survival in two clinical cohorts of 238 patients with HGSC (left, log rank. p = 0.037) and 23 patients with oropharyngeal squamous cell carcinomas (right, log. rank p = 0.027), respectively.

Tissue samples were divided into smaller pieces and treated with 0, 2 or 8 Gy irradiation, fixed in formalin after 1h and paraffin embedded. After tissue processing, the samples were investigated for IGF1R/PCNA colocalization using PLA and γ-H2AX reactivity using immunohistochemistry. Quantification for IGF1R/PCNA PLA signals was carried out by manually counting 100 tumor cells in at least three different randomly selected areas of the tumor. For comparison between cases and treatments, we calculated the fraction of "positive" cells ($\geq$3 PLA signals per cell based on the number of signals per cell, determined by the median number of signals per cell across all experiments). In all cases both 2 Gy and 8 Gy X-ray increased IGF1R/PCNA colocalization (Fig 4B) and formations of γH2AX foci (S8 Fig).

We visually compared the staining pattern of adjacent normal tissue (normal epithelial tissue, adipose tissue and stromal cells). As compared to the tumor, normal cells with low proliferation showed formation of γH2AX but no IGF1R/PCNA colocalization. Similarly, the tumor cell tissues with higher proliferation (such as colon epithelium) showed induction of both γH2AX and IGF1R/PCNA. This suggests that IGF1R/PCNA colocalization may be a strategy for cells to maintain proliferation in the presence of DNA damage.

## IGF1R/PCNA interaction is absent in most cancer cell lines

Since IGF1R/PCNA interaction was associated with better outcome in the two clinical cancer cohorts, we sought to identify its functional role in cell lines. To identify which cell lines had an active IGF1R/PCNA interaction, co-immunoprecipitation (co-IP) experiments were

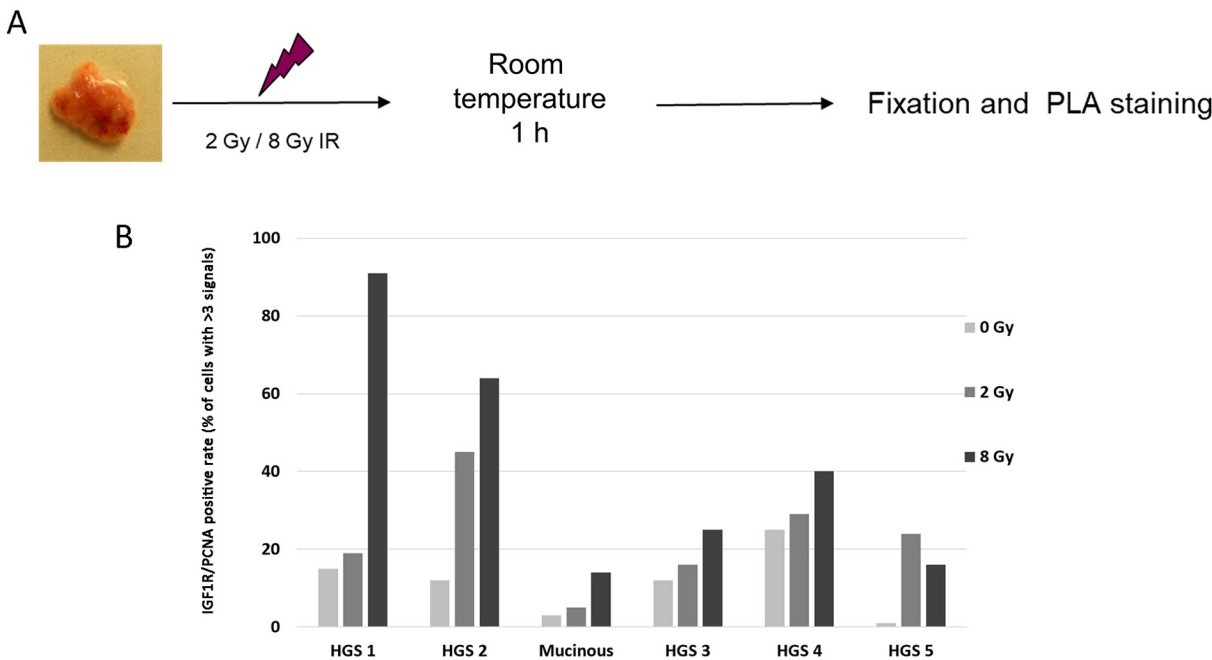

**Fig 4. Ex vivo irradiation of clinical tumors induces IGF1R/PCNA interaction in ovarian cancer.** A) Schematic experimental design of ex vivo irradiation (see Material and methods 2.5). B) Nuclear IGF1R/PCNA PLA signals were calculated for 100 individual cells in at least four high-power fields. The height of columns in the top panel represents the fraction of tumor cells with ≥ 3 signals for IGF1R/PCNA foci per cell. There was a significant inverse correlation between IGF1R/PCNA levels at 8 Gy.

performed on a panel of established cancer cell lines (Fig 5A and S9 Fig). While most cancer cell lines expressed IGF1R at varying levels (Fig 5B), the only cell lines which repeatedly showed positive co-immunoprecipitation at basal conditions were HeLa cells and IGF1R-over-expressing MEF cell line cells (R+). Subsequent fluorescent PLA showed that the colocalization of IGF1R and PCNA was predominantly nuclear in HeLa cells (Fig 5C). To elucidate the function of IGF1R/PCNA interaction in HeLa cells, we used hydroxyurea (HU) to induce DNA replication stress/replication fork stalling. Though no change in co-IP protein levels was detected in IGF1R after HU treatment, PCNA was both mono-ubiquitinated and co-immuno-precipitated with the E3 ubiquitin ligase Rad18 and HLTF (Fig 5D). These effects were inhibited by IGF1R kinase inhibitor NVP suggesting that this effect is dependent on IGF1R kinase activity. These results are in line with our previous mechanistic study using an R+ cell line and other human non-malignant cell lines which showed that nuclear IGF1R kinase activity is essential for PCNA ubiquitination and TLS. However, when expression levels of TLS DNA polymerases (S11A Fig) and the foci formation of DNA polymerase ETA (POLH) (S12 Fig) were inspected after 8 Gy irradiation with or without IGF1/NVP treatment in HeLa cells, no significant change was observed. The expression of POLH was not altered by irradiation in either of the *ex vivo* samples as indicated by IHC experiments (S11B Fig).

## Nuclear IGF1R activation rescues stalling DNA replication forks in HeLa cells

Next, we wanted to investigate if the IGF1R/PCNA interaction was associated with DDT/replication fork stress. The ability of IGF1R to rescue stalling replication forks was studied by measuring DNA fibers after combinations of HU, IGF1 and NVP treatments in co-IP positive

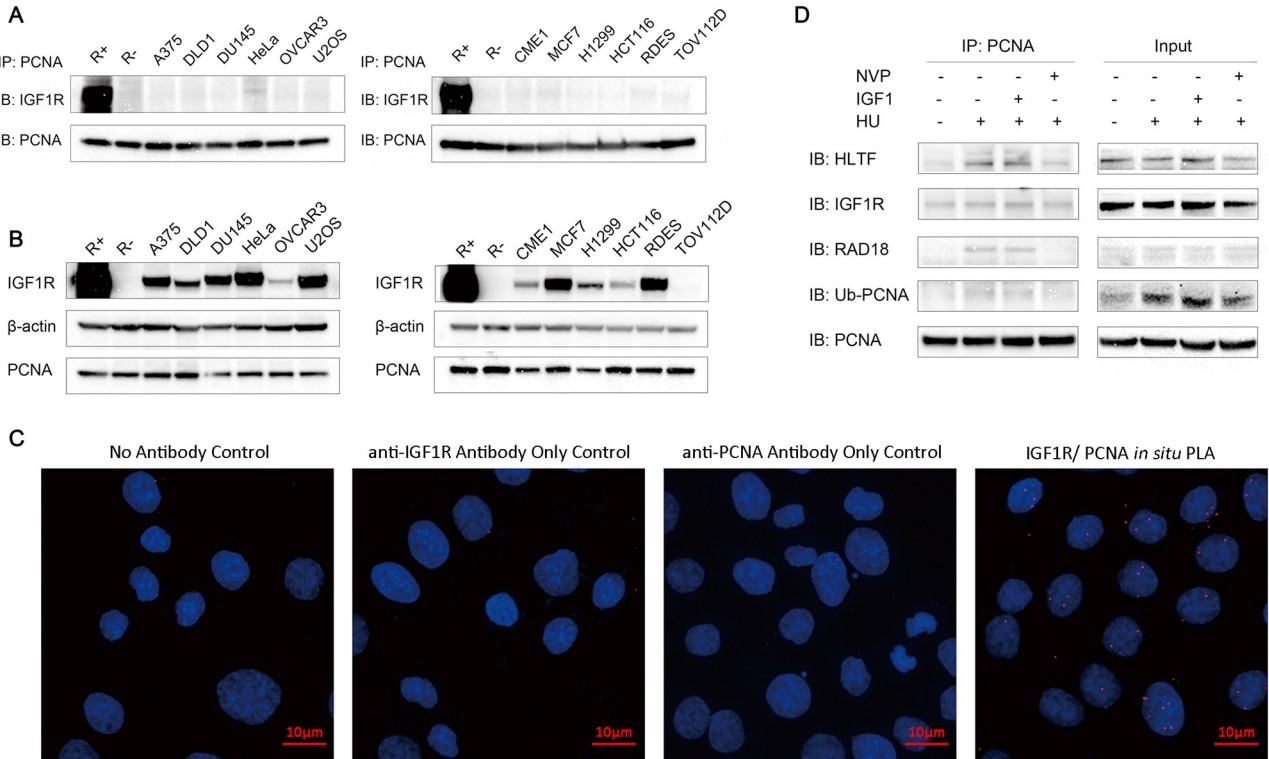

**Fig 5. nIGF1R colocalizes with PCNA in HeLa cells, but rarely in other cancer cell lines.** A) PCNA was pulled down from a panel of cancer cell lines by immunoprecipitation (IP). The co-immunoprecipitation of IGF1R with PCNA was analyzed through immunoblotting of IGF1R after SDS-PAGE separation. The membranes were reblotted for PCNA to control that the IP was successful. R+ and R- cells were included as positive and negative controls. B) Immunoblotting showing the various expression levels of IGF1R and PCNA in the co-IP input samples of the cancer cell lines. R+ and R- cells were used as controls. β-actin was blotted to control the equal loading. C) Colocalization of PCNA and IGF1R in Hela cells was visualized by immunofluorescent *in situ* PLA. Red foci indicate IGF-1R/PCNA interactions. Counterstaining with DAPI (blue) shows cell nuclei. The negative controls were obtained by omitting one or both of the primary antibodies. D) HeLa cells were treated with hydroxyurea (2mM) for 1h with or without IGF1 (50ng/ml) and NVP (1μM) before immunoprecipitation with anti-PCNA. The co-immunoprecipitation of HLTF, IGF1R, RAD18 and ubiquitin-PCNA was detected through immunoblotting. The total cell lysate samples of each condition were stained as input controls.

HeLa and R+ cells and co-IP negative HCT116, SW480 and R- cells (as shown above). DNA fork stalling was effectively induced by 0.2 mM HU treatment, represented by the decrease of DNA elongation ratio after and before HU treatment compared to untreated cells (Fig 6 and S10 Fig). IGF1 treatment significantly increased replication fork restart after HU-mediated fork stalling in HeLa and R+ cells, but not in HCT116 (IGF1R negative), SW480 (IGF1R positive) or R- cells, indicating that this effect is dependent on both the kinase activity of IGF1R and its interaction with PCNA. Similarly, the ameliorating effect of IGF1 in HeLa and R+ was also effectively inhibited by IGF1R inhibition by NVP. Thus, IGF1R mediates rescuing of stalled replication forks, and is dependent on interaction with PCNA and its kinase activity.

## Discussion

There is growing evidence that nuclear translocation of receptor tyrosine kinases (RTKs) in cancer cells functions as a resistance mechanism to increase tolerance for conventional and targeted therapy [20, 21]. Early work by us and others has identified the nuclear trafficking mechanisms of IGF1R [22–24]. Since nuclear IGF1R has been linked to an aggressive clinical course and therapy resistance in different types of cancer, much IGF1R related research

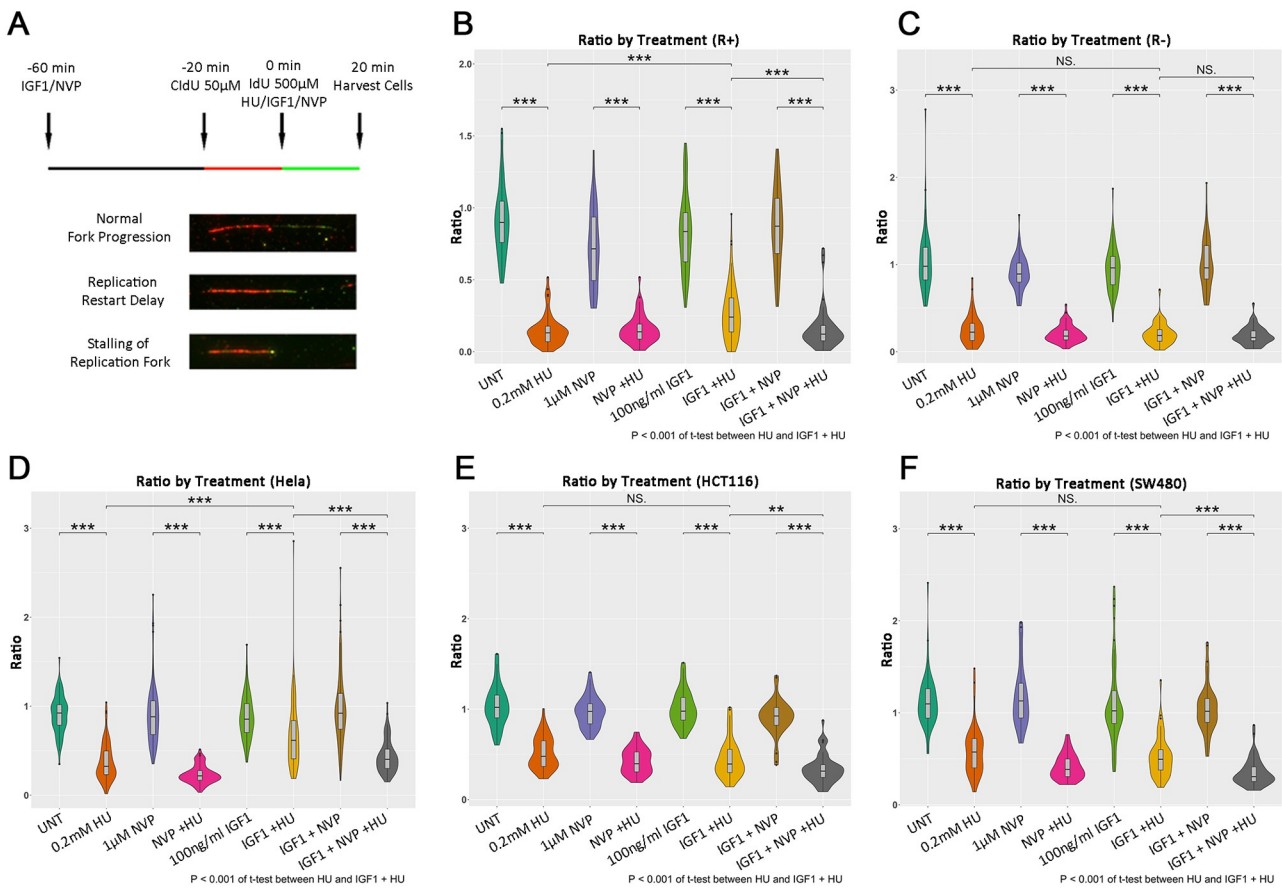

**Fig 6. IGF1R rescue stalled DNA replication forks in cervical cancer cells.** nIGF1R involved in restart of DNA replication fork stalling induced by Hydroxyurea (HU). The upper panel of (A) shows a schematic figure that illustrates the experiment procedure. At 60 min before HU treatment, 100ng/ml IGF1, 1μM NVP or 100ng/ml IGF + 1μM NVP were added to different treatment groups. After incubating for 40 min, 50μM CIdU (red) was added to all treatment groups in order to pulse-label the DNA for 20 min. Cells were then washed twice by warm medium and labeled again with medium with 500μM IdU (Green). The treatments of 0.2mM HU as well as the IGF1, NVP and IGF1 + NVP in different groups were supplemented in the mediums synchronously as indicated (B-F). After incubating for another 20 min, samples were harvested by trypsin digestion for further analyses. The lower panel of (A) shows typical staining appearances of fork events as detected by DNA fiber assay. In (B) R+ cells, (C) R- cells, (D) HeLa, (E) HCT116 and (F) SW480 replication restart speed was represented by the ratio of the lengths of nascent replication tracts after inducing DNA damage labeled with IdU (green) to the length of the replication tract before inducing damage labeled with CIdU (red). At least 50 unidirectional forks labeled with both CIdU and IdU were measured for each condition.

currently focuses on nuclear IGF1R. Our recent discovery that IGF1R could increase cellular tolerance to DNA replication stress by direct phosphorylation of PCNA in stem cells prompted us to investigate this mechanism in cancer cells.

In the process of carcinogenesis, cancer cells must alter basic mechanisms to cope with replication stress including alteration of DNA repair pathways. Accumulated knowledge of these mechanisms and their mutual relationships have been fundamental in understanding how human cancer can develop and also allowed clinicians to target specific pathways in cancer, e.g. using PARP inhibitors in BRCA-deficient cancer. The mechanisms governing DNA damage tolerance and their association with DNA-repair pathways is still incompletely understood. As argued below, we believe that the IGF1R/PCNA interaction is a basic cellular mechanism to increase tolerance to DNA stress during proliferation, but that this mechanism is lost with tumor progression in conjunction with accumulated DNA damage and aberrant strategies to tolerate genomic instability.

In this work we show that IGF1R interacts with PCNA after irradiation and HU treatment in HeLa cells (cervical cancer cells) and IGF1R positive fibroblasts. IGF1R kinase inhibition abolished PCNA/RAD18 interaction and PCNA-mono-ubiquitination showing that this mechanism extends to cancer cells. Activation of IGF1R rescued HU-induced stalled replication forks, but this effect was only found in cell lines with detectable IGF1R/PCNA interaction. Across different types of cancer tissues, we observed that nuclear IGF1R was colocalized with PCNA, indicating that a significant proportion of human cancers depend on this mechanism for DNA damage tolerance. Indeed, IGF1R/PCNA colocalization in individual tumors was stronger in areas which are associated with higher proliferation (tumor invasive front) and genomic instability (dysplastic areas and cells with greater nuclear atypia). The colocalization was also found in highly proliferative normal tissues. Unexpectedly, the IGF1R/PCNA colocalization was absent in most metastatic lesions, neoadjuvantly treated tumors (with poor treatment response) and most cancer cell lines. We also found a significant correlation between IGF1R/PCNA and RAD18/PCNA colocalization in cancer tissue, indicating a possible link to TLS activation. This correlation was not seen in metastatic lesions, suggesting that other mechanisms may activate TLS independent of IGF1R.

By investigating two independent tumor cohorts of HGSC and OPSCC, respectively, we found that strong IGF1R/PCNA colocalization was significantly associated with better overall survival. This difference could be explained by the association between nuclear IGF1R and HR DNA repair, and this hypothesis has been supported by numerous previous studies; i) the known association between HR deficiency and increased survival in these two tumor groups (by means of BRCAness/treatment responsiveness) [25–27], ii) IGF1R inhibition decreases HR DNA repair [28], iii) IGF1R and BRCA1 share a common function in the way they phosphorylate PCNA [29]; and iv) inhibition of BRCA1 makes cells more susceptible to IGF1R inhibition [12]. As described by Dr. Heyer and others, the rescue of stalled replication forks by the DNA damage tolerance pathway is closely related and probably interlinked with the HR DNA repair [6, 30]. In a series of *ex vivo* irradiated ovarian carcinomas, we found that both low and high dosage of X-ray increased IGF1R/PCNA colocalization together with an increase in γH2AX foci. This strongly implies that IGF1R/PCNA interaction is acutely upregulated after induction of double strand breaks and should be linked to the rescue of stalled replication forks.

Using PLA, we frequently observed varying levels of cytoplasmic IGF1R/PCNA signals. While the number of cytoplasmic signals seemed to correlate with nuclear signals, we are currently unable to explain the nature of this colocalization. Speculatively, these signals could be associated with cellular co-transport and will be subject to future investigation.

The varying presence of IGF1R/PCNA colocalization in clinical tumors suggests that a significant proportion of clinical cases depend on IGF1R as a strategy to tolerate DNA damage. HR deficiency has been coupled to increased TLS, which may result in increased cisplatin resistance in ovarian cancer and head and neck squamous cell carcinomas [31, 32] which could explain the association with overall survival in our clinical cohorts. This may indicate that IGF1R-targeting strategies could benefit from additive effects with irradiation, platinum or PARP-based treatment–especially earlier in the clinical course of cancer.

## Supporting information

**S1 Text. Material and methods.**
(DOCX)

**S1 Fig.**
(JPG)

**S2 Fig.**
(JPG)

**S3 Fig.**
(JPG)

**S4 Fig.**
(JPG)

**S5 Fig.**
(JPG)

**S6 Fig.**
(JPG)

**S7 Fig.**
(JPG)

**S8 Fig.**
(JPG)

**S9 Fig.**
(TIF)

**S10 Fig.**
(JPG)

**S11 Fig.**
(JPG)

**S12 Fig.**
(JPG)

**S1 Table. Proximity ligation assay (PLA) in clinical cases.**
(XLSX)

## Acknowledgments

The authors would like to thank Professor Arne Östman for valuable scientific discussion and the clinical staff of the Department of Clinical Pathology and Cytology at the Karolinska University Hospital for helping with the collection of clinical samples.

## Author Contributions

**Conceptualization:** Olle Larsson, Felix Haglund.

**Data curation:** Sara Corvigno, Hanna Dahlstrand, Joseph Carlson, Anders Näsman.

**Formal analysis:** Chen Yang, Mingzhi Liu, Yingbo Lin, Olle Larsson, Felix Haglund.

**Funding acquisition:** Olle Larsson.

**Investigation:** Chen Yang, Yifan Zhang, Yi Chen, Franziska Ragaller, Anders Näsman, Felix Haglund.

**Methodology:** Ahmed Waraky, Yingbo Lin, Olle Larsson, Felix Haglund.

**Project administration:** Olle Larsson, Felix Haglund.

**Resources:** Zihua Chen, Anders Näsman.

**Software:** Mingzhi Liu.

**Supervision:** Olle Larsson, Felix Haglund.

**Visualization:** Chen Yang, Yingbo Lin, Felix Haglund.

**Writing – original draft:** Chen Yang, Felix Haglund.

**Writing – review & editing:** Yifan Zhang, Yingbo Lin, Olle Larsson.

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
