## [Decision Letter · Decision Letter 0]

1 Apr 2020

PONE-D-20-06136

Nuclear IGF1R interact with PCNA to preserve DNA replication after DNA-damage in a variety of human cancers

PLOS ONE

Dear %Dr. Haglund%,

Thank you for submitting your manuscript to PLOS ONE. After careful consideration, we feel that it has merit but does not fully meet PLOS ONE’s publication criteria as it currently stands. Therefore, we invite you to submit a revised version of the manuscript that addresses the points raised during the review process.

We would appreciate receiving your revised manuscript by May 16 2020 11:59PM. To enhance the reproducibility of your results, we recommend that if applicable you deposit your laboratory protocols in protocols.io, where a protocol can be assigned its own identifier (DOI) such that it can be cited independently in the future. For instructions see: http://journals.plos.org/plosone/s/submission-guidelines#loc-laboratory-protocols

We look forward to receiving your revised manuscript.

Kind regards,

Komaraiah Palle, Ph.D.

Academic Editor

PLOS ONE

Journal Requirements:

Additional Editor Comments (if provided):

Reviewers' comments:

Reviewer's Responses to Questions

**Comments to the Author**

1. Is the manuscript technically sound, and do the data support the conclusions?

Reviewer #1: Yes

Reviewer #2: Partly

2. Has the statistical analysis been performed appropriately and rigorously? 

Reviewer #1: Yes

Reviewer #2: Yes

3. Have the authors made all data underlying the findings in their manuscript fully available?

Reviewer #1: Yes

Reviewer #2: Yes

4. Is the manuscript presented in an intelligible fashion and written in standard English?

Reviewer #1: Yes

Reviewer #2: No

5. Review Comments to the Author

Reviewer #1: The manuscript focuses on the interaction between IGF1R and PCNA in various tumors and cancer cells lines, and its association with overall survival. Most of the tumors exhibited IGF1R/PCNA colocalizations. Furthermore, this type of colocalization is minimized in tumors with poor treatment response, as well as in metastatic lesions. The manuscript is written well, however, few revisions should be made to improve the overall quality.

Minor revisions:

1. In DNA Fiber assay, CidU should be changed to CldU throughout the manuscript, and in figures.

2. Scale in µm should be included in all figures with microscopy images.

3. Figures 4 and 5: fgi should be changed to fig

4. The quality of all figures appear to be low and the text within the figures is not legible. The figures should be at least 300dpi. Especially in figure 6, the images and graphs are very hard to see.

5. In figure 5, the signals for IGF1R are too saturated. For the text “Though no change in co-IP protein levels was detect in IGF1R after HU treatment, PCNA was both mono-ubiquitinated and co-immunoprecipitated with the E3 ubiquitin ligase Rad18 and HLTF (Figure 5C),” the corresponding figure should be changed to “Figure 5D”.

6. “Subsequent fluorescent PLA showed that the colocalization of IGF1R and PCNA was predominantly nuclear in HeLa cells (Figure 5D).” The figure should be “Figure 5C” here, which corresponds to immunofluorescence. In this figure, color for IGF1R and PCNA should be different to clearly visualized for colocalization.

Reviewer #2: In this manuscript, authors evaluated the role of nuclear IGF1R and its correlation with poor outcomes in cancer. Mechanistically, in situ proximity ligation assay identified frequent IGF1R and PCNA co-localization across many cancer types. While IGF1R/PCNA co-localization was found to be increased frequently in tumor cells versus adjacent normal tissue as well as in areas with dysplasia and invasion, the interaction was frequently lost in tumors with poor response to neo-adjuvant treatment and most metastatic lesions. Additionally, in clinical samples of serous ovarian carcinomas and oropharyngeal squamous cell carcinomas, stronger IGF1R/PCNA co-localization was significantly associated with better overall survival. IGF1R activation also rescues stalled DNA replication forks, but only in cancer cells with baseline IGF1R/PCNA interaction. Overall, authors summarize that cancer cells may utilize IGF1R phosphorylation of PCNA to increase DNA damage tolerance, but this mechanism is frequently lost with tumor progression.

• The manuscript clearly lacks clarity and mostly confusing. A clear explanation of the results and its significance in discussion is needed in the revised manuscript.

• There are numerous spelling mistakes in the manuscript. Authors have to completely re-check for these small error. Few are highlighted below.

• Line 69: PCNA

• Line119: for in

• Line 147: IGF1R

• Include scale bars in figure 1 and 2.

• Conclusion for this manuscript is achieved mostly by PLA assay. Authors should use additional methods or ways to confirm their result. More validation of TLS pathways is needed to confirm the conclusion.

• Apart from the nuclear PLA signal between IGF1R and PCNA, there are also some signals in cytoplasm. It will be helpful for the readers if authors can discuss about the cytoplasmic PLA signal’s significance.

• Line 153 to 163 corresponds to which figure? Supplementary figure 1 to 5 is not cited in the results section.

6. PLOS authors have the option to publish the peer review history of their article (what does this mean?). If published, this will include your full peer review and any attached files.

Reviewer #1: No

Reviewer #2: No

---

## [Author Response · Author response to Decision Letter 0]

23 Jun 2020

Dear Dr. Heber, 

Thank you for considering our manuscript entitled “Nuclear IGF1R interact with PCNA to preserve DNA replication after DNA-damage in a variety of human cancers” (PONE-D-20-06136) for publication in PLOS ONE. We much appreciate the reviewers’ optimistic feedback and suggestions which we have addressed in a point-by-point manner to the best of our ability. 

Despite the current pandemic situation that has put significant constraints to our work environment, we have performed experiments to strengthen the soundness of the manuscript. Please note that the addition of new experimental data has resulted in two additional supplementary figures and texts within the manuscript. We have also uploaded the uncropped western blot graphs as well as other raw data that relate to this manuscript to public available database with a DOI: 10.6084/m9.figshare.12388256.

Felix Haglund, MD / PhD

Consultant Pathologist and Associate Professor of Experimental Pathology

Department of Oncology-Pathology

Karolinska Institutet and

Karolinska University Hospital 

S-171 76 Stockholm, Sweden 

Work Phone: +46851770000

Fax: +46851776180

 

Reviewer #1: 

The manuscript focuses on the interaction between IGF1R and PCNA in various tumors and cancer cells lines, and its association with overall survival. Most of the tumors exhibited IGF1R/PCNA colocalizations. Furthermore, this type of colocalization is minimized in tumors with poor treatment response, as well as in metastatic lesions. The manuscript is written well, however, few revisions should be made to improve the overall quality.

Minor revisions:

Issue: “1. In DNA Fiber assay, CidU should be changed to CldU throughout the manuscript, and in figures.”

Response: The abbreviation has been corrected as suggested.

Issue: “2. Scale in µm should be included in all figures with microscopy images.”

Response: Scales have been added to all microscopy images as suggested.

Issue: “3. Figures 4 and 5: fgi should be changed to fig”

Response: The mistake has been corrected. 

Issue: “4. The quality of all figures appear to be low and the text within the figures is not legible. The figures should be at least 300dpi. Especially in figure 6, the images and graphs are very hard to see.”

Response: New high-quality images have been produced as suggested.

Issue: “5. In figure 5, the signals for IGF1R are too saturated. For the text “Though no change in co-IP protein levels was detect in IGF1R after HU treatment, PCNA was both mono-ubiquitinated and co-immunoprecipitated with the E3 ubiquitin ligase Rad18 and HLTF (Figure 5C),” the corresponding figure should be changed to “Figure 5D”.”

Response: i) The error with figure citation has been corrected. ii) The R+ is a cell line with IGF1R overexpression. To get the IGF1R signal in the cancer cell lines on the same membrane at visible level, the R+ signal will get overexposed when loading similar equivalent levels of protein. 

Issue: “6. “Subsequent fluorescent PLA showed that the colocalization of IGF1R and PCNA was predominantly nuclear in HeLa cells (Figure 5D).” The figure should be “Figure 5C” here, which corresponds to immunofluorescence. In this figure, color for IGF1R and PCNA should be different to clearly visualized for colocalization.”

Response: i) The error with figure citation has been corrected and colors adjusted as suggested. 

ii) We apologize that the nature of Figure 5C has been unclear. The red signals represent fluorescent proximity ligation assay (PLA) signals and are the product of juxtaposition of two antibodies (anti-IGF1R and anti-PCNA). We understand that these figures look like the more common immunofluorescence figures and have added in situ PLA to each photograph of Figure 5C for clarification. 

We have also expanded the section on PLA in in the Material and Methods section:

In situ Proximity Ligation Assay (PLA)

See Supplementary Material and Methods. Representative microphotographs of IGF1R/PCNA PLA validation are shown in Supplementary Figure 4 (FFPE tissue, brown dots indicating colocalization of the two proteins) and Figure 5 D (Cells, red fluorescence dots indicating colocalization of the two proteins).

 

Reviewer #2: 

In this manuscript, authors evaluated the role of nuclear IGF1R and its correlation with poor outcomes in cancer. Mechanistically, in situ proximity ligation assay identified frequent IGF1R and PCNA co-localization across many cancer types. While IGF1R/PCNA co-localization was found to be increased frequently in tumor cells versus adjacent normal tissue as well as in areas with dysplasia and invasion, the interaction was frequently lost in tumors with poor response to neo-adjuvant treatment and most metastatic lesions. Additionally, in clinical samples of serous ovarian carcinomas and oropharyngeal squamous cell carcinomas, stronger IGF1R/PCNA co-localization was significantly associated with better overall survival. IGF1R activation also rescues stalled DNA replication forks, but only in cancer cells with baseline IGF1R/PCNA interaction. Overall, authors summarize that cancer cells may utilize IGF1R phosphorylation of PCNA to increase DNA damage tolerance, but this mechanism is frequently lost with tumor progression.

Issue: “• The manuscript clearly lacks clarity and mostly confusing. A clear explanation of the results and its significance in discussion is needed in the revised manuscript.”

Response: As suggested, we have added a paragraph discussing the significance of our results. We have also reorganized parts of the discussion to make it easier to follow our arguments.

The added paragraph in the Discussion:

“In the process of carcinogenesis, cancer cells must alter basic mechanisms to cope with replication stress including alteration of DNA repair pathways. Accumulated knowledge of these mechanisms and their mutual relationships have been fundamental in understanding how human cancer can develop and also allowed clinicians to target specific pathways in cancer, e.g. using PARP inhibitors in BRCA-deficient cancer. The mechanisms governing DNA damage tolerance and their association with DNA-repair pathways is still incompletely understood. As argued below, we believe that the IGF1R/PCNA interaction is a basic cellular mechanism to increase tolerance to DNA stress during proliferation, but that this mechanism is lost with tumor progression in conjunction with accumulated DNA damage and aberrant strategies to tolerate genomic instability.”

Issue: “• There are numerous spelling mistakes in the manuscript. Authors have to completely re-check for these small error. Few are highlighted below.

• Line 69: PCNA

• Line119: for in

• Line 147: IGF1R

• Include scale bars in figure 1 and 2.”

Response: i) Scale bars have been introduced as suggested. ii) We’ve had the whole manuscript proof read by a native-English speaker and small changes have been made accordingly throughout the manuscript.

Issue: “• Conclusion for this manuscript is achieved mostly by PLA assay. Authors should use additional methods or ways to confirm their result. More validation of TLS pathways is needed to confirm the conclusion.”

Response: We agree that further investigation of the translesion synthesis (TLS) pathway is interesting. During additional experiments, we have measured the gene expression levels of DNA polymerases involved in TLS in HeLa cells after irradiation (8 Gy) and IGF-1R modulators (IGF1/NVP). We were unable to detect any significant differences (Supplementary Figure 11A), indicating that IGF-1R signaling does not directly affect the gene expression of these polymerases. Similarly, we were unable to detect a difference in protein expression of DNA polymerase eta (POLH) in ovarian cancer tissue that was irradiated ex vivo. These data have been added as Supplementary Figure 11. Previous results showed that POLH formed nuclear foci in HeLa cells exposed to irradiation, indicating that irradiation activates TLS in HeLa cells. We stained the POLH foci using an immunofluorescence-based method, but no significant difference was induced by the IGF-1R modulators (IGF1/ NVP). These data have been added as Supplementary Figure 12.

Co-immunoprecipitation experiments on HeLa cells under replication stress (Figure 5D), show that the IGF-1R inhibitor NVP abolished the PCNA interaction with the E3 ligase Rad18 and HLTF. These experiments show that Rad18-mediated activation of TLS is dependent on IGF1R kinase activity, a notion further supported by the effects of NVP in the replication fork experiments (Figure 6D). We have added an additional text to the Abstract and Discussion emphasizing the functional implications of these experiments:

The added text in the Abstract:

“In vitro, RAD18 mediated mono-ubiquitination of PCNA during replication stress was dependent on IGF1R kinase activity.”

The added paragraph in the Discussion:

“In this work we show that IGF1R interacts with PCNA after irradiation and HU treatment in HeLa cells (cervical cancer cells) and IGF1R positive fibroblasts. IGF1R kinase inhibition abolished PCNA/RAD18 interaction and PCNA-mono-ubiquitination showing that this mechanism extends to cancer cells. Activation of IGF1R rescued HU-induced stalled replication forks, but this effect was only found in cell lines with detectable IGF1R/PCNA interaction.”

Issue: “• Apart from the nuclear PLA signal between IGF1R and PCNA, there are also some signals in cytoplasm. It will be helpful for the readers if authors can discuss about the cytoplasmic PLA signal’s significance.”

Response: In principle there are two different explanations for the cytoplasmic signals. First, the signals could be technical artifacts related to local precipitations of the two antibodies, perhaps related to high prevalence of cytoplasmic PCNA. However, R- cells are expressing high levels of PCNA but completely lack IGF1R-PCNA PLA. Furthermore, from the stained cancer tissues, we could not see a clear correlation between overall IGF1R or PCNA immunoreactivity and the level of PLA-signals (i.e. tumors expressing both proteins could be negative on PLA). In situ PLA signals are supposed to be quite specific in their localization which would argue against some kind of “spill-over” effect from the nuclear signals. The other explanation is that the signals are real, and reflect a true IGF1R-PCNA colocalization in the cytoplasm of the cell. While IGF-1R has well established functions in the cell membrane little is known about cytoplasmic PCNA. We’re unfortunately currently unable to show any mechanistic function of this cytoplasmic interaction. We’ve done some investigative experiments, including cytoplasmic co-IP, and believe that this interaction is real. However, we thought that this investigation was outside the scope of this paper and hope to be able to present this investigation in a separate publication. As suggested, we have added a paragraph to the discussion:

“Using PLA, we frequently observed varying levels of cytoplasmic IGF1R/PCNA signals. While the number of cytoplasmic signals seemed to correlate with nuclear signals, we are currently unable to explain the nature of this colocalization. Speculatively, these signals could be associated with cellular co-transport and will be subject to future investigation.”

Issue: “• Line 153 to 163 corresponds to which figure? Supplementary figure 1 to 5 is not cited in the results section.”

Response: i) The section relates to our findings shown in Supplementary Table 1, which is now correctly cited at the end of the paragraph. ii) As suggested, we have added an additional paragraph to the Results section citing the Supplementary Figures. 

“Prior to investigating tumor specimens, the specificity and concentrations of the utilized antibodies were validated on cells and tissue with known expression of the target proteins (Supplementary Figure 1-2). The in-situ PLA method was validated in tissue samples with confirmed immunoreactivity (Supplementary Figure 3 and 4) as described in the Supplementary Material and Methods.”

---

## [Decision Letter · Decision Letter 1]

6 Jul 2020

Nuclear IGF1R interact with PCNA to preserve DNA replication after DNA-damage in a variety of human cancers

PONE-D-20-06136R1

Dear Dr. %Haglund%,

We’re pleased to inform you that your manuscript has been judged scientifically suitable for publication and will be formally accepted for publication once it meets all outstanding technical requirements. However, there are some typographical errors that needs to be addressed before publication. Please see below for the reviewer's comments.

Kind regards,

Komaraiah Palle, Ph.D.

Academic Editor

PLOS ONE

Additional Editor Comments (optional):

Although minor revisions pertaining to spellings and grammar were edited well throughout the text, following typographical errors were observed and need to be edited.

1. In the Supplementary Table 1, Spelling of ‘skeletal’ in 18th row to be corrected.

2. In the Supplementary Information on Materials and Methods, 17th line, ‘the slides was…’ to be corrected to ‘the slides were…’

3. In the Supplementary Figure 10, ‘CidU’ to be corrected to ‘CIdU’

4. In the Supplementary Figure 12, the words, ‘significant and treatments’ are wrongly spelled.

5. The word DNA polymerase eta is mentioned differently (ETA, Eta and eta in text and figures). The notation should be uniform throughout the manuscript.

Overall, the clarity and flow of content is better in the revised draft. The manuscript is as per the standards of the Journal and hence can be accepted after verifying and editing the minor errors.

Reviewers' comments:

Reviewer's Responses to Questions

**Comments to the Author**

1. If the authors have adequately addressed your comments raised in a previous round of review and you feel that this manuscript is now acceptable for publication, you may indicate that here to bypass the “Comments to the Author” section, enter your conflict of interest statement in the “Confidential to Editor” section, and submit your "Accept" recommendation.

Reviewer #3: All comments have been addressed

2. Is the manuscript technically sound, and do the data support the conclusions?

Reviewer #3: Yes

3. Has the statistical analysis been performed appropriately and rigorously? 

Reviewer #3: Yes

4. Have the authors made all data underlying the findings in their manuscript fully available?

Reviewer #3: Yes

5. Is the manuscript presented in an intelligible fashion and written in standard English?

Reviewer #3: Yes

6. Review Comments to the Author

Reviewer #3: Review comments on the revised manuscript (PONE-D-20-06136R1)

submitted to PLOS ONE Journal

In the revised manuscript entitled, “Nuclear IGF1R interact with PCNA to preserve DNA replication after DNA-damage in a variety of human cancers” the author(s) have incorporated the changes appropriately as per the suggestions of the reviewers. The text is thoroughly revised to improvise results and discussion sections while citing or editing the figure and table numbers at the relevant places. The authors have attempted to incorporate additional experimental results and emphasized on elaborating the significance of the results obtained in the discussion as suggested by one of the reviewers. The resolution and format of the figures has also been enhanced as commented.

Although minor revisions pertaining to spellings and grammar were edited well throughout the text, following typographical errors were observed and need to be edited.

1. In the Supplementary Table 1, Spelling of ‘skeletal’ in 18th row to be corrected.

2. In the Supplementary Information on Materials and Methods, 17th line, ‘the slides was…’ to be corrected to ‘the slides were…’

3. In the Supplementary Figure 10, ‘CidU’ to be corrected to ‘CIdU’

4. In the Supplementary Figure 12, the words, ‘significant and treatments’ are wrongly spelled.

5. The word DNA polymerase eta is mentioned differently (ETA, Eta and eta in text and figures). The notation should be uniform throughout the manuscript.

Overall, the clarity and flow of content is better in the revised draft. The manuscript is as per the standards of the Journal and hence can be accepted after verifying and editing the minor errors.

7. PLOS authors have the option to publish the peer review history of their article (what does this mean?). If published, this will include your full peer review and any attached files.

Reviewer #3: No

---

## [Editor Report · Acceptance letter]

10 Jul 2020

PONE-D-20-06136R1 

Nuclear IGF1R interact with PCNA to preserve DNA replication after DNA-damage in a variety of human cancers 

Dear Dr. Haglund:

I'm pleased to inform you that your manuscript has been deemed suitable for publication in PLOS ONE. Congratulations! Your manuscript is now with our production department. 

Kind regards, 

on behalf of

Dr. Komaraiah Palle 

Academic Editor

PLOS ONE